# Analytical and clinical validation of a multiplex PCR assay for detection of *Neisseria gonorrhoeae* and *Chlamydia trachomatis* including simultaneous LGV serotyping on an automated high-throughput PCR system

Lisa Sophie Pflüger,[1] Dominik Nörz,[1] Moritz Grunwald,[1] Susanne Pfefferle,[1] Katja Giersch,[1] Martin Christner,[1] Beatrice Weber,[1] Martin Aepfelbacher,[1] Holger Rohde,[1,2] Marc Lütgehetmann[1,2]

**ABSTRACT** For effective infection control measures for *Chlamydia trachomatis* (*CT*) and *Neisseria gonorrhoeae* (*NG*), a reliable tool for screening and diagnosis is essential. Here, we aimed to establish and validate a multiplex PCR assay on an automated system using a dual-target approach for the detection of *CT/NG* and differentiation between lymphogranuloma venereum (LGV) and non-LGV from genital and extra-genital specimens. Published primer/probe sets (*CT*: pmpH, cryptic plasmid; *NG*: porA, opa) were modified for the cobas 5800/6800/8800. Standards quantified by digital PCR were used to determine linearity and lower limit of detection (LLoD; eSwab, urine). For clinical validation, prospective samples (*n* = 319) were compared with a CE-marked *in vitro* diagnostics (CE-IVD) assay. LLoDs ranged from 21.8 to 244 digital copies (dcp)/mL and 10.8 to 277 dcp/mL in swab and urine, respectively. A simple linear regression analysis yielded slopes ranging from −4.338 to −2.834 and Pearson correlation coefficients from 0.956 to 0.994. Inter- and intra-run variability was <0.5 and <1 cycle threshold (ct), respectively. No cross-reactivity was observed (*n* = 42). Clinical validation showed a sensitivity of 94.74% (95% confidence interval (CI): 87.23%–97.93%) and 95.51% (95% CI: 89.01%–98.24%), a specificity of 99.59% (95% CI: 97.71%–99.98%) and 99.57% (95% CI: 97.58%–99.98%), positive predictive values of 89.91% (estimated prevalence: 3.7%; 95% CI: 80.91%–95.6%) and 88.61% (estimated prevalence: 3.4%; 95% CI: 80.18%–94.34%), and negative predictive values of 99.81% (95% CI: 98.14%–100%) and 99.85% (95% CI: 98.14%–100%) for the detection of *CT* and *NG*, respectively. In conclusion, we established a dual-target, internally controlled PCR on an automated system for the detectiwon of *CT/NG* from genital and extra-genital specimens. Depending on local regulations, the assay can be used as a screening or a confirmatory/typing assay.

**IMPORTANCE** *Chlamydia trachomatis* (*CT*) and *Neisseria gonorrhoeae* (*NG*) represent a major global health burden, with the World Health Organization estimating that >128 million and >82 million people, respectively, were newly infected in 2020. For effective infection control measures, a reliable tool for sensitive diagnosis and screening of *CT/NG* is essential. We established a multiplex PCR assay for the detection of *CT/NG* and simultaneous discrimination between lymphogranuloma venereum (LGV) and non-LGV strains, which has been validated for genital and extra-genital specimens on a fully automated system. To increase assay sensitivity, a dual-target approach has been chosen for both pathogens. This strategy reduces false-positive results in oropharyngeal swabs due to the detection of commensal *N.* species that may harbor *NG* DNA fragments targeted in the PCR due to horizontal gene transmission following previous

Address correspondence to Marc Lütgehetmann, mluetgeh@uke.de.

Marc Lütgehetmann and Dominik Nörz received travel expenses and speakers' honoraria from Roche (Rotkreuz, Switzerland) and Qiagen (Hilden, Germany). The other authors declare no conflicts of interest.

See the funding table on p. 12.

infection. In sum, the established assay provides a powerful tool for use as either a screening/diagnostic or a typing/confirmatory assay.

**KEYWORDS**  multiplex PCR, high throughput, dual-target PCR, sexually transmitted disease, *Chlamydia trachomatis*, lymphogranuloma venereum, *Neisseria gonorrhoeae*, LGV serotyping

Sexually transmitted diseases (STDs) represent a major global health burden. The World Health Organization estimates that over 1 million new sexually transmitted infections (STIs) occur every day worldwide (1). Two of the most common STDs are infections caused by *Chlamydia trachomatis* (CT) and *Neisseria gonorrhoeae* (NG) with an estimated number of new infections of 128 and 82 million in 2020, respectively (1). Particularly in women, most gonococcal or chlamydial infections are asymptomatic or have minimal symptoms. This results in undiagnosed and untreated patients, which increases the risk of transmission and leads to high numbers of unreported STI cases (2). Furthermore, infections with CT and NG can have serious sequelae that include a negative impact on fertility and the development of long-term complications, especially if infections are not detected and patients are left untreated. Both CT and NG are readily treatable with antibiotics. Unfortunately, the amount of NG strains that are antimicrobial-resistant (AMR) is rising, posing a progressive threat to global health (3–9). Consequently, a rapid and reliable method for screening and diagnosis of the respective STIs is essential. The most accurate method for detecting symptomatic and asymptomatic infections with NG or CT is a nucleic acid amplification test (NAAT). NAATs are the recommended method for all clinical specimens by the Center for Disease Control and Prevention (10, 11). As symptoms of both STDs are similar and coinfection of the two pathogens is common, testing of CT should also prompt testing for NG and vice versa (12, 13).

In addition, CT serovar typing is crucial as the duration of antibiotic treatment depends on the serovar present. A total of 19 different CT serovars are known (A-L3). Infections with serovars L1-L3 cause lymphogranuloma venereum (LGV), which can be invasive, causing systemic infection and potentially irreversible sequelae if treatment is not started promptly. Thus, CT serovar typing is essential, particularly in high-risk groups (e.g., men who have sex with men).

Both CT and NG can cause extra-genital infections including proctitis and pharyngitis. In high-risk groups such as men who have sex with men and patients with a history of oral or anal sex, extra-genital CT and/or NG infections are not uncommon and are often asymptomatic, stressing the need for screening from extra-genital specimens (10, 14, 15). However, the detection of STIs from pharyngeal swabs poses a particular challenge for NAATs, as *non-NG Neisseria* species (*N.* spp.), which can be part of the physiological oropharyngeal flora, may harbor fragments of NG following previous infections, thereby generating false-positive results (16, 17). To increase specificity, especially in non-genital swabs, a dual-target approach has been proven advantageous to ensure reliable test results (18, 19).

The aim of this study was to compile a multiplex PCR assay for the detection of CT and NG that simultaneously differentiates between CT serovar A-K (non-LGV strains) and L1-L3 (LGV strains) by applying a dual-target approach for both pathogens. The laboratory-developed test (LDT) PCR was adapted and validated for use on a fully automated system, the cobas 5800/6800/6800, thus providing a rapid and reliable method for the detection of CT and NG from oropharyngeal, anorectal, cervical/vaginal, urethral/penile, and ocular swabs as well as urine samples.

## RESULTS

### *In silico* analysis

No critical oligo–oligo interaction could be identified, indicating that the overall risk of primer dimer formation is minimal (cutoff binding energy: delta G < −10 kcal/mol). However, one 309-base pair (bp)-long human DNA sequence on chromosome 7 was identified, which can potentially be amplified by one of the primers used (opa-rev_01; all oligonucleotide names mentioned in this manuscript correspond to those in Table 1). The assay performance showed no impairment due to increased primer consumption, and alignments revealed five mismatches between the respective DNA sequence and the opa-rev_01 with one to two mismatches being located within the last 10 bases from the 3′ end of the primer.

### Technical validation

The 95% lower limits of detection (LLoDs) were established using an eight-step dilution series with 21 repeats per dilution step for two different matrixes (swab and urine, Table 2). LLoD (eSwab medium) for the assay detecting all *CT* serovars was determined to be 32.7 dcp (digital copies: copy number determined by digital PCR)/mL [95% confidence interval (CI): 23.3–56.1 dcp/mL] and 60.9 dcp/mL (95% CI: 41.8–108 dcp/mL) for the LGV-specific assay. The assay targeting the opa gene showed a LLoD of 21.8 dcp/mL (95% CI: 15.2–38.3 dcp/mL), and for the assay amplifying a PCR product within the porA pseudogene, a LLoD of 244 dcp/mL (95% CI: 141–705 dcp/mL) was determined. Using pooled urine that tested negative for *CT* and *NG* by PCR, all assays demonstrated excellent LLoDs. Only the assay targeting the porA pseudogene performed slightly better in eSwab media with a higher LLoD of 277 dcp/mL (95% CI: 186–485 dcp/mL) in urine compared to 244 dcp/mL (95% CI: 141–705 dcp/mL) in the eSwab medium. To determine whether LLoDs in swabs collected from different body sites are similar to those determined in the eSwab medium, pooled clinical specimens (oropharyngeal, anal/rectal, cervical/vaginal, urethral/penile, and ocular swabs) that tested negative for *CT* and *NG* by PCR were used (six-step dilution series with eight repeats per dilution step). Oropharyngeal and ocular swabs showed comparable performance, whereas for anal/rectal and urethral/penile swabs, the LLoDs were slightly higher. Cervical/vaginal swabs, however, performed significantly worse compared to the other matrixes tested (Table S1).

All assays showed excellent linearity (variation <1 ct). For *CT* serovars A-L3 and L1-L3, a linear range was observed from 30.1 to 40.7 and 28.6 to 42.0 ct with slopes of −3.509 and −4.338 and Pearson correlation coefficient ($r^2$) of 0.994 and 0.968, respectively. The second-/third-order polynomial showed the best fit for both assays detecting *NG*. However, the coefficients were not statistically significant. Simple linear regression revealed a slope of −3.272 and −2.834 and $r^2$ of 0.992 and 0.956 for the assays targeting the opa and porA gene, within the measuring interval of 27.4–40.1 and 26.7–38.3 ct, respectively (Fig. 1).

Inter- and intra-run variability surpassed the set goal of less than 0.5 and 1 ct for each assay. Within-run and between-day variability ranged from 0.0473 to 0.148 ct and 0.0163 to 0.137 ct, respectively, demonstrating a high reproducibility of test results (detailed results of precision experiments are listed in Table S2). No cross-reactivity was observed using a set of different viral and bacterial pathogens ($n$ = 42, Table S3). All external quality assessment (EQA) samples ($n$ = 26) were identified correctly by the Utility Channel assay for the detection of *CT* and *NG* (UC_CTNG, $n$ = 18/18 positive and $n$ = 8/8 negative for *CT*; $n$ = 9/9 positive and $n$ = 6/6 negative for *NG*).

### Clinical validation

The UC_CTNG assay is considered *NG*-positive if both the opa and porA targets are reported as positive in this manuscript. If discrepant results are observed (opa+/porA− or opa−/porA+), either the test must be performed again or confirmation of *NG* by culture is

**TABLE 1** Assay design: for adaption on the cobas 5800/6800/8800, primers were modified including a 2′O-methyl-RNA base (indicated as 2′0-Meth-X) to prevent the formation of primer cross- and self-dimers[a,b]

| Pathogen | Target | Oligo name | Sequence 5′ to 3′ | Channel | Conc. (nM) | Reference |
|---|---|---|---|---|---|---|
| *Chlamydia trachomatis* | plasmid | plsmd-fwd | GGA TTG ACT CCG ACA ACG TAT (2′0-Meth-U) C | | 500 | 20 |
| | | plsmd-rev | ATC ATT GCC ATT AGA AAG GGC A (2′0-Meth-U) T | | 500 | |
| | | nonLGV-probe_01 | SUN- TTA CGT GTA (BMN-Q535) GGC GGT TTA GAA AGC GG -BMN-Q535 | 3 | 100 | |
| | pmpH | pmpH-fwd | GGA TAA CTC TGT GGG GTA TTC TC (2′0-Meth-C) T | | 500 | |
| | | pmpH-rev | AGA CCC TTT CCG AGC ATC A (2′0-Meth-C)T | | 500 | |
| | | nonLGV-probe_02 | SUN- GCT TGA AGC (BMN-Q535) AGC AGG AGC TGG TG -BMN-Q535 | 3 | 100 | |
| | | LGV-probe | FAM- T[+C]C GCT TG[+C] TCC A[+A]C AGT -BHQ1 | 2 | 100 | this study |
| *Neisseria gonorrhoeae* | opa | opa-fwd | TTG AAA CAC CGC CCG (2′0-Meth-G) AA | | 500 | 21 |
| | | opa-rev_01 | TTT CGG CTC CTT ATT CGG TT (2′0-Meth-U) AA | | 500 | |
| | | opa-rev_02 | TTT CGG CTC CTT ATT CGG TTT (2′0-Meth-G) A | | 500 | |
| | | opa-probe | Atto425- CCG ATA TAA TC[+C] GYC [+C]TT CAA [+C]AT CAG -BHQ1 | 1 | 100 | |
| | porA | porA-fwd | CAG CAT TCA ATT TGT TCC GAG (2′0-Meth-U) C | | 500 | 22, 23 |
| | | porA-rev | GAA CTG GTT TCA TCT GAT TAC TTT C (2′0-Meth-C) A | | 500 | |
| | | porA-probe | Atto620- CGC CTA TAC (BHQ2) GCC TGC TAC TTT CAC GC -BHQ2 | 4 | 100 | |

[a]Locked nucleic acids are indicated as "[+X]" and were used to enhance assay performance by increasing the hybridization melting temperature. The LGV-specific probe was modified from Cole et al. (5) and supplemented by four additional bases at the 5′ end. The concentrations correspond to the final concentration of the respective oligonucleotide in the reaction mix.

[b]Conc., concentration; BHQ, black hole quencher; BMN-Q, BMN-quencher; fwd, forward; LNA, locked nucleic acid; MGB, minor groove binder; opa, opacity protein; plsmd, plasmid; pmpH, polymorphic membrane protein H; rev, reverse.

warranted (Fig. 2). A comparison of assay performance to a CE-IVD assay (cobas *CT/NG*) as a reference standard revealed a sensitivity of 94.74% (95% CI: 87.23%–97.93%) and 95.51% (95% CI: 89.01%–98.24%) and a specificity of 99.59% (95% CI: 97.71%–99.98%) and 99.57% (95% CI: 97.58%–99.98%) for the assays detecting *CT* and *NG*, respectively. To calculate the positive and negative predictive values, the prevalence for both pathogens was calculated based on a retrospective analysis of 4,298 genital and extra-genital specimens tested at our center (1 November 2022–31 May 2023, as part of this study) and was found to be 3.7% and 3.4% for *CT* and *NG*, respectively. Positive predictive values were determined to be 89.91% (95% CI: 80.91%–95.6%) and 88.61% (95% CI: 80.18%–94.34%) and negative predictive values as 99.81% (95% CI: 98.14%–100%) and 99.85% (95% CI: 98.14%–100%) for the detection of *CT* and *NG*, respectively.

The quantitative correlation analysis showed a strong correlation between the assays with an $r^2$ of 0.9034 (target: pmpH non-LGV-specific and cryptic plasmid), 0.9283 (target: pmpH LGV-specific), 0.9862 (target: opa), and 0.9794 (target: porA; Fig. 3). To analyze the agreement between the UC_CTNG and the CE-IVD assay, a Bland–Altman comparison was conducted. The analysis revealed a mean of the bias of 1.359 [95% limits of agreement (±1.96 standard deviations): 4.436 and −1.718], −2.393 (95% limits of agreement: −0.969 and −3.818), and −2.02 (95% limits of agreement: −0.2959 and −3.745) for comparison of the CE-IVD and LDT assay detecting *CT* (all serovars), *NG* opa target, and *NG* porA target, respectively (Fig. S1). For the assays detecting *CT* (all serovars) and the detection of *NG* (both UC_CTNG *NG* targets positive), one false-positive (UC_CTNG: positive; CE-IVD assay: negative) and four false-negative (UC_CTNG: negative; CE-IVD assay: positive) test results were identified each (see Fig. 2). False-negative results occurred exclusively in samples with high ct values (>37.13 and >39.58 ct for the assays detecting *CT* and *NG*, respectively) and, thus, estimated low DNA load. Similarly, false-positive results occurred in patient samples with estimated low target concentrations (*NG*: opa target: 36.9 ct and porA target: 35.5 ct; *CT*: non-LGV-specific targets: 40.83 ct; Table S4).

## Assay performance in routine diagnostics

A total of 4,298 samples consisting of swabs (n = 2,876), urine (n = 1,391), and assorted/unassigned samples (n = 31) were screened with the UC_CTNG assay between 1

TABLE 2 Lower limit of detection: standards were quantified with digital PCR (dPCR) and used for the determination of lower limit of detection by creating an eight-step dilution series with $n$ = 21 repeats per dilution step[c,d]

| | | Chlamydia trachomatis | | | | Neisseria gonorrhoeae | | | |
| | | serovar A-L3[a] | | serovar L1-L3[a] | | opa target[b] | | porA target[b] | |
| | Conc. (dcp/mL) | Positive results | Hit rate | Positive results | Hit rate | Positive results | Hit rate | Positive results | Hit rate |
|---|---|---|---|---|---|---|---|---|---|
| Swab | 400 | 21/21 | 100% | 21/21 | 100% | n/a | n/a | n/a | n/a |
| | 200 | 21/21 | 100% | 21/21 | 100% | 21/21 | 100% | 20/21 | 95.24% |
| | 100 | 21/21 | 100% | 21/21 | 100% | 21/21 | 100% | 17/21 | 80.95% |
| | 50 | 21/21 | 100% | 20/21 | 95.24% | 21/21 | 100% | 14/21 | 66.67% |
| | 25 | 19/21 | 90.5% | 17/21 | 80.95% | 21/21 | 100% | 12/21 | 57.14% |
| | 12.5 | 16/21 | 76.2% | 10/21 | 47.62% | 21/21 | 100% | 6/21 | 28.6% |
| | 6.25 | 5/21 | 23.8% | 4/21 | 19.05% | 15/21 | 71.43% | 0/21 | 0% |
| | 3.13 | 6/21 | 28.6% | 6/21 | 28.57% | 17/21 | 80.95% | 0/21 | 0% |
| | 1.56 | n/a | n/a | n/a | n/a | 7/21 | 33.33% | 0/21 | 0% |
| | Overall (dcp/mL) | Established LLoD | 95% CI | Established LLoD | 95% CI | Established LLoD | 95% CI | Established LLoD | 95% CI |
| | | 32.7 | 23.3–56.1 | 60.9 | 41.8–108 | 21.08 | 15.2–38.3 | 244 | 141–705 |
| Urine | 1,000 | 21/21 | 100% | 21/21 | 100% | 21/21 | 100% | 21/21 | 100% |
| | 500 | 21/21 | 100% | 21/21 | 100% | 21/21 | 100% | 21/21 | 100% |
| | 250 | 21/21 | 100% | 21/21 | 100% | 21/21 | 100% | 20/21 | 100% |
| | 125 | 21/21 | 100% | 21/21 | 100% | 21/21 | 100% | 21/21 | 100% |
| | 62.5 | 21/21 | 100% | 21/21 | 100% | 21/21 | 100% | 21/21 | 100% |
| | 31.25 | 21/21 | 100% | 20/21 | 95.24% | 21/21 | 100% | 4/21 | 19.05% |
| | 15.63 | 21/21 | 100% | 19/21 | 90.48% | 20/21 | 95.24% | 9/21 | 42.86% |
| | 7.81 | 17/21 | 80.95% | 9/21 | 42.86% | 18/21 | 85.71% | 2/21 | 9.52% |
| | 3.91 | 7/21 | 33.33% | 9/21 | 42.86% | 14/18 | 66.67% | 1/21 | 4.76% |
| | Overall (dcp/mL) | Established LLoD | 95% CI | Established LLoD | 95% CI | Established LLoD | 95% CI | Established LLoD | 95% CI |
| | | 10.8 | 8.55–17.4 | 29 | 20.3–53.8 | 13.4 | 9.34–31.9 | 277 | 186–485 |

[a]Targeted gene for dPCR quantification: pmpH (single-copy gene).
[b]Targeted gene for dPCR quantification: opa (multi-copy gene).
[c]n/a, not applicable; PCR, polymerase chain reaction.
[d]Either swab matrix and cobas PCR media (Roche, Rotkreuz, Switzerland) or urine that tested negative for CT and NG by PCR and cobas PCR media (ratio: 50:50) was used for dilution. Concentrations represent digital copies per milliliter specimen.

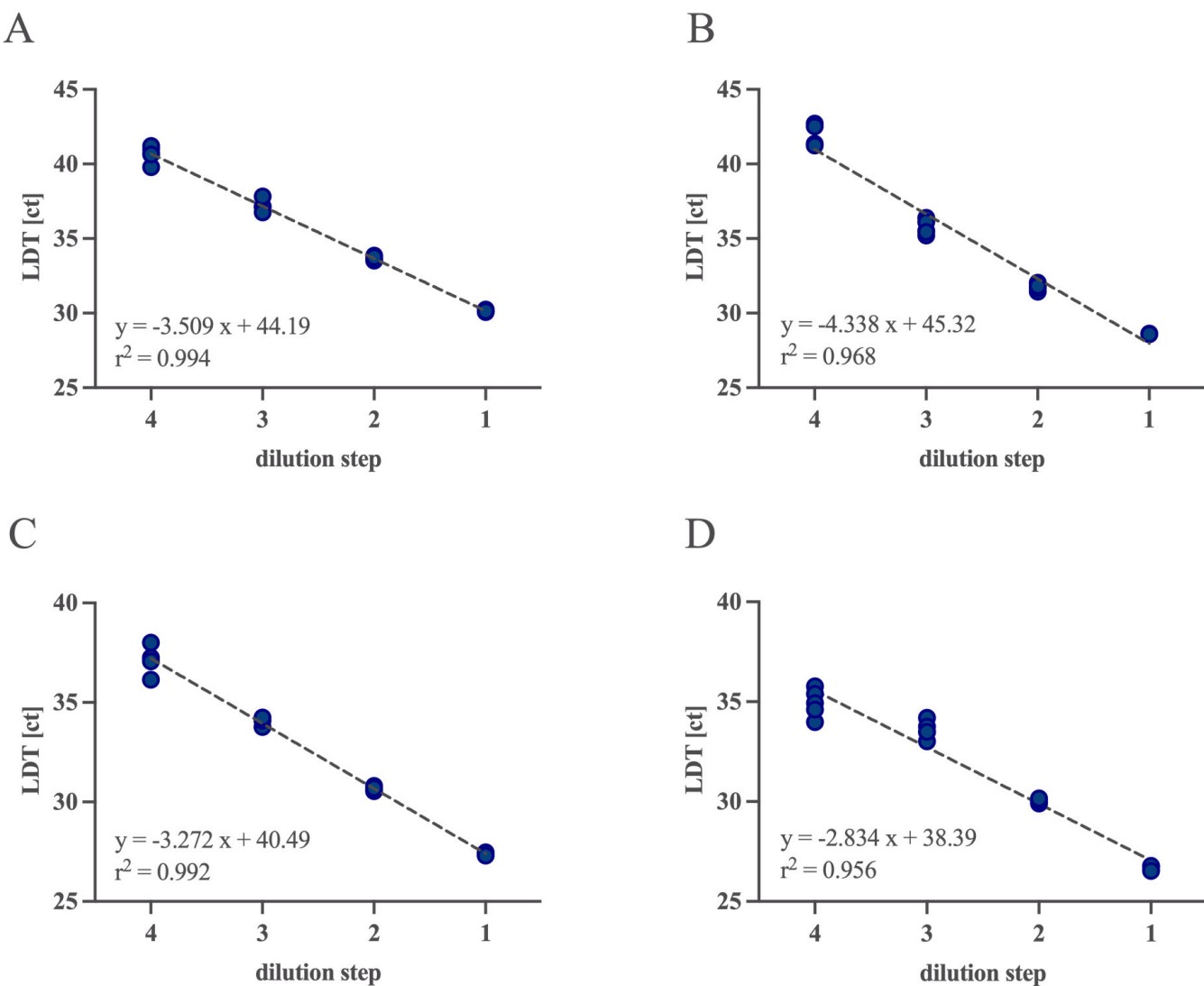

**FIG 1** Linearity: to assess linearity, standards were quantified using dPCR from clinical samples. A 10-fold dilution series (four steps, $n$ = 5 repeats/ dilution) was created and subjected to the UC_CTNG assay on the same device and day. A simple linear regression analysis demonstrated slopes ranging from −2.834 to −4.338 and a Pearson correlation coefficient ($R^2$) ranging from 0.956 to 0.994. Plotted are the linear regression lines (dashed gray lines) and ct test results (blue dots) for the assays detecting (A) *CT* serovars A-L3, (B) *CT* serovars L1-L3, (C) *NG* (target: opa), and (D) *NG* (target: porA). Abbreviation: UC_CTNG; Utility Channel assay for detection of *Chlamydia trachomatis* and *Neisseria gonorrhoeae*.

November 2022 and 31 May 2023, of which 1,962 were additionally tested by *NG* culture. The majority of the screened swabs consisted of oropharyngeal ($n$ = 1,174) and anal/ rectal ($n$ = 1,085) swabs; also, urethral/penile ($n$ = 325), vaginal/cervical ($n$ = 250), ocular ($n$ = 26), and unassigned ($n$ = 16) swabs were tested. In total, 162/4,298 *CT*-positive samples were identified including 8/162 LGV-positive samples, and in 145/4,298 samples, *NG* was detected (opa and porA target reactive). Of 4,298 samples, 28 were positive for both pathogens, including three LGV strains. In 4.5% of chlamydial infections, LGV strains were identified (Table S5). No culture-positive *NG* infection was missed with the UC_CTNG assay. In 53/54 *NG* culture-positive samples (98.2%), both *NG* targets were detected, and in 1/54 (1.8%; conjunctival swab), only the opa tested positive (opa: 20 ct, porA: non-reactive). Further, 62 samples were tested positive for both *NG* targets with the molecular assay but remained negative in culture. Analysis of all oropharyngeal swabs ($n$ = 1,174/4,298) that were tested during the respective time period identified 15 samples that tested positive for one *NG* target only (opa positive: 13/15, mean: 34.32 ct; range: 24–41 ct; porA positive: 2/15, mean ct: 36.4; ct values: 35.2 and 37.6). None of these

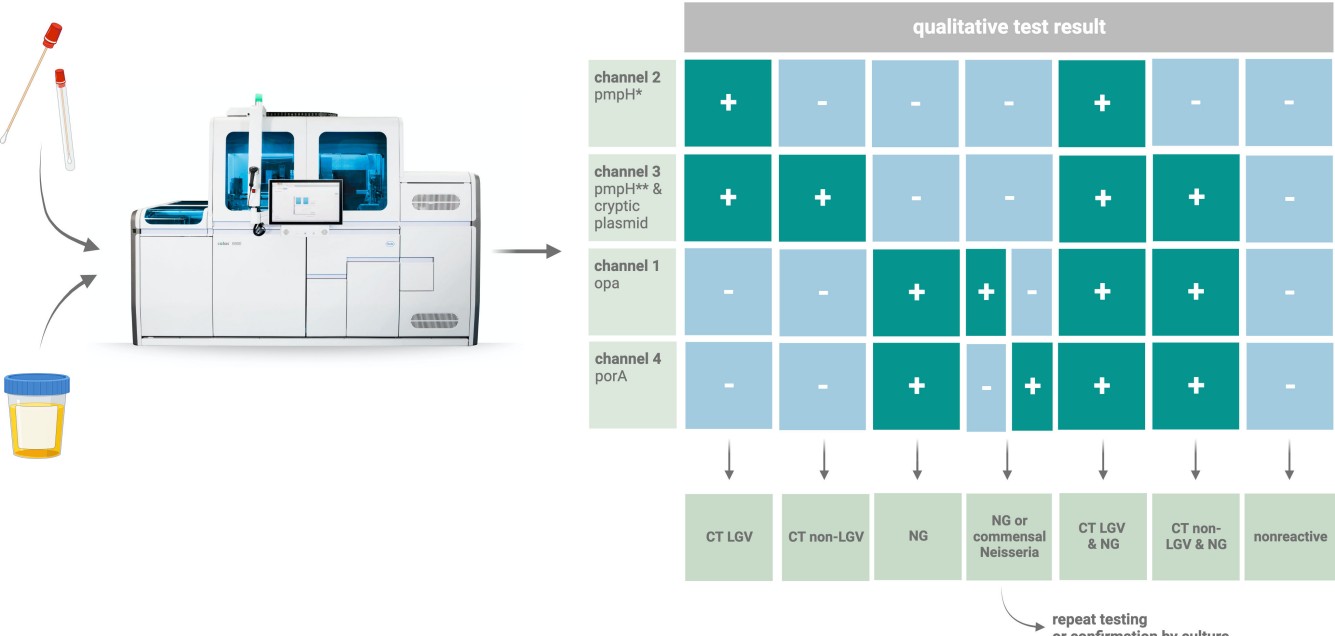

**FIG 2** Interpretation of test results: if only one of the two *NG* targets is amplified, it is recommended that the test result should be confirmed by culture or re-tested within a reasonable time, as commensal *N.* spp. could encode for one of the target genes due to horizontal gene transmission (HGT). If the respective internal control (IC) is negative (not shown; detection in channel 5), the test results are invalid and need to be repeated. The figure was created using BioRender.com. *Detection of the LGV-specific target (the probe hybridizes to the junction area of the deletion region within the pmpH gene that is strictly present in LGV strains). **Detection of the non-LGV-specific target within the pmpH gene. Abbreviation: +, reactive; −, non-reactive.

samples showed growth of *NG* in culture. However, in 4/15 (26.67%) samples, *Neisseria meningitides* was later identified by culture (mean: 31.6 ct, range: 24–38.6 ct; Table S6), suggesting that horizontal transmission of the PCR target gene had occurred between the two species.

## DISCUSSION

We established a dual-target PCR assay for the detection of *NG*, *CT,* and LGV typing on a high-throughputCR system featuring a full-process control. This design approach increases the reliability of detection coverage in case of mutations/deletions within the target regions, as *CT* strains that present a 377-bp-long deletion area within the cryptic plasmid sequence and plasmid-free *CT* isolates as well as PorA-negative *NG* isolates have been reported (16, 17, 24–28).

A dual-target approach is essential to avoid false-positive results from oropharyngeal swabs due to the detection of commensal *N.* spp. The oropharyngeal mucosa represents an ideal niche for *N.* spp., where *NG* and *N. meningitides* can coexist among commensal *N.* spp., a situation favorable for HGT (29). Thus, commensal *N.* spp. may encode for one of the genes targeted in thePCR assay (30–33). Consequently, we recommend that samples tested positive for only one of the *NG* targets should be confirmed by culture or re-evaluated within a reasonable time frame. The results of our retrospective analysis of 4,298 samples as part of the evaluation of the clinical performance of the UC_CTNG assay support this practice, as they show that discrepant test results may occur and can be resolved using this two-line strategy. This approach is particularly important for screening in high-risk groups for oropharyngeal infections, as oropharyngeal swabs are the second most common material after anal swabs in which *NG* can be detected (33.3% vs 46.43%; retrospective analysis of $n = 2,690$ samples from patients who visited our center's post-exposure consultation within 7 months).

Moreover, an assay that helps guide the discrimination of gonococcal infections from commensal *N.* spp. is important since HGT is playing a major role in the increased

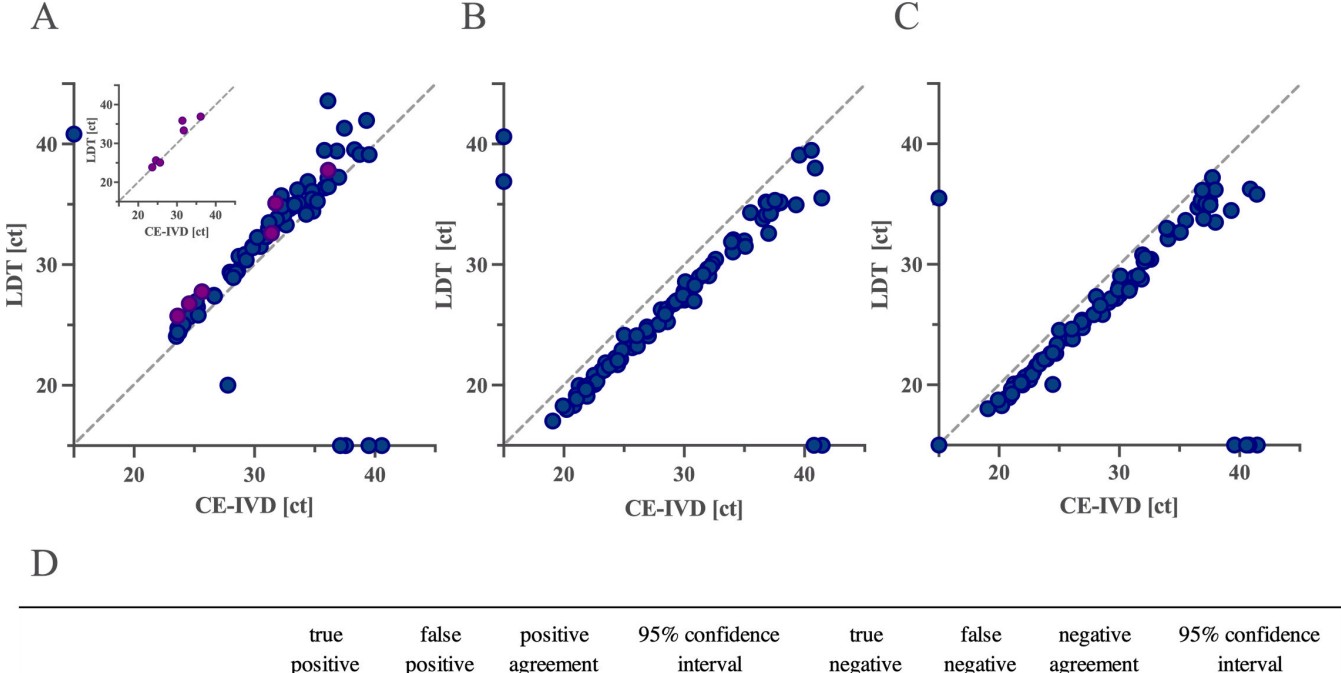

| | true positive | false positive | positive agreement | 95% confidence interval | true negative | false negative | negative agreement | 95% confidence interval |
|---|---|---|---|---|---|---|---|---|
| **Chlamydia trachomatis** | 76 | 1 | 94.74% | 87.23% - 97.93% | 243 | 4 | 99.59% | 97.71% - 99.98% |
| **Neisseria gonorrhoeae** | 85 | 1 | 95.51% | 89.01% - 98.24% | 229 | 4 | 99.57% | 97.58% - 99.98% |

**FIG 3** Correlation: correlation of test results (ct values) of the LDT (UC_CTNG) and CE-IVD (CT/NG, Roche, Rotkreuz, Switzerland) assays detecting (A) *CT* serovars A-L3 (LGV strains are plotted as violet dots), (B) *NG* (LDT target: opa), and (C) *NG* (LDT target: porA). In the upper left-hand corner of (A), test results of the LDT detecting LGV strains are plotted compared to the CE-IVD assay detecting all *CT* serovars. All samples were tested on the same device and day. Samples that tested negative for one of the assays are plotted on the respective axis. The table shows an overall agreement between the UC_CTNG and CT/NG assay that was used as a reference standard. All false-negative test results occurred with low positive samples (all ct >37). Ct values for false-negative and false-positive test results are displayed in Table S5. Samples that tested positive for *NG* for only one target (opa or porA) with the LDT assay were considered negative.

number of AMR *NG*. Variations in the penA gene that are associated with cephalosporin resistance have been reported to be from commensal origin, underlining the clinical importance and implications arising from HGT as well as the need for consequent screening strategies (30, 34, 35). Oropharyngeal *NG* infections are frequently asymptomatic, resulting in undiagnosed cases due to insufficient testing, which in turn promotes disease transmission and dissemination. Therefore, an adequate screening regime and a reliable diagnostic tool are of utmost importance.

However, while reliably detecting all *NG* strains, our assay cannot predict the presence of AMR *NG*. This limitation warrants a culture approach or second-line PCR testing using recently published primer/probe sets targeting *NG* genes associated with AMR (36–38). This approach is particularly relevant in case of treatment failure, as the incidence of AMR *NG* is steadily rising (39, 40).

In addition, the here-described assay is able to differentiate between LGV and non-LGV *CT* strains without the need for additional PCR reactions. LLoDs for non-LGV and LGV *CT* strains differed by less than factor 2 [tested in two different matrixes that represent typical clinical specimens: viral transport media (eSwab) and urine] enabling *CT* serovar typing even in low positive samples, which was previously shown to be challenging (25). Of all *CT*-infected patients in our cohort, 4.5% (8/162) were LGV-positive. Although the overall prevalence in our mixed population was lower than the 15%–16.5% found by a study in German high-risk populations (men who have sex with men) (41), our data highlight the importance of differentiating between LGV and non-LGV to allow optimal patient care.

The here-described validation of the UC_CTNG assay was performed in accordance with the current European guidelines (regulation 2017/746 EU IVDR). However,

requirements and guidelines for validation and verification of LDTs for diagnostic purposes vary from country to country as well as between regulatory bodies. Therefore, verification experiments are warranted, especially for the different matrixes (incl. different swab collection systems) for which the assay will be used, if the UC_CTNG assay is implemented in another laboratory for diagnostic purposes.

In conclusion, we established a dual-target internally controlled LDT PCR assay on the open channel of a fully automated system for molecular diagnostic of *CT/NG* and simultaneous LGV typing from genital and extra-genital specimens. The assay was highly sensitive and demonstrated comparative assay performance to a competitor assay (CE-IVD marked and FDA cleared) in our clinical set. The dual-target principle of the UC_CTNG assay can reduce false-positive test results in oropharyngeal swabs due to the detection of commensal *N*. spp. and enables sensitive differentiation between LGV/ non-LGV *CT* strains at the same time. Depending on local regulations, the assay could be used as either a screening or a confirmatory typing assay run on a fully automated platform.

## MATERIALS AND METHODS

### Assay design and PCR setup

Primers/probe sets from previously published diagnostic assays were selected and adapted for use on the cobas 5800/6800/8800 to enable the simultaneous detection of non-LGV, LGV *CT*, and *NG* strains in one reaction (21–23, 42, 43). For the detection of *CT*, primers and probes targeting the pmpH and cryptic plasmid gene as described by Chen et al. were chosen (43). To identify LGV strains (serovars L1-L3), two different probes are used, each hybridizing within the pmpH gene but labeled with a different fluorophore. This concept relies on the presence of a 36-bp deletion in the pmpH gene, which is only present in LGV strains (44). The FAM-labeled probe (LGV-probe) hybridizes to a LGV-specific sequence (junction area) whereas the SUN-labeled probe (nonLGV-probe_02) hybridizes to a conserved region within the deletion area, hence detecting *CT* serovars D-K (Fig. 4) (42). The second *CT* PCR target is the multi-copy cryptic plasmid gene. The PCR product is 87 bp long, and the corresponding probe (nonLGV-probe_01) detects all *CT* serovars. Both non-LGV-specific probes (nonLGV-probe_01 and nonLGV-probe_02) are labeled with the same fluorophore (SUN), thus generating a fluorescence signal in the same channel (channel 3). For the detection of *NG*, a primer/probe set targeting the PorA pseudogene on account of its highly conserved sequence and specificity for *NG* is used (22, 23). The second *NG* target is the multi-copy opa gene, a highly conserved region of the *NG* genome (45). The PCR products are 89 and 90 bp long, respectively. The assay is regarded as *NG*-positive if both *NG* targets yield a positive test result. Discrepant test results of the two *NG* targets require confirmation through culture or re-testing. The interpretation of test results of the multiplex assay is shown in Fig. 2.

The sequences of the utilized oligonucleotides are listed in Table 1. For adaptation on the cobas 5800/6800/8800 system, primers and probes were modified accordingly. Modifications included the use of locked nucleic acids to increase hybridization melting temperature, thereby improving binding (27, 28). To further increase $T_m$, the LGV-specific probe (LGV-probe) was supplemented by four additional bases at the 5′ end compared to its original published sequence by Morré et al. (42). Next, long sequence probes were double-quenched to reduce background fluorescence. Furthermore, a second reverse primer (opa-rev_02) targeting the opa gene was added, which differs only in the penultimate base to cover a critical mismatch at this position. Consequently, the 2′O-methylation had to be changed in this primer to the base before the penultimate base. All primers and probes were ordered from and custom-made by Ella Biotech (Fürstenfeld-brock, Germany).

As multiplex PCR assays harbor the risk of formation of primer dimers resulting in primer consumption and unspecific amplification of human DNA/RNA due to the use of a variety of different oligonucleotides, all utilized primers and probes were analyzed using

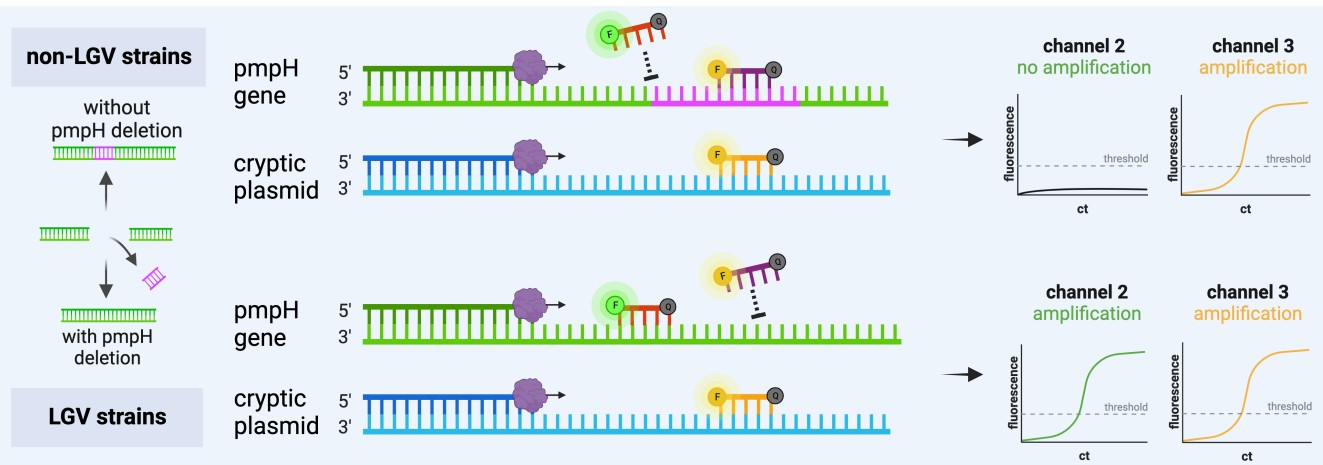

**FIG 4** Discrimination between LGV and non-LGV *CT* strains: A dual-target approach was chosen for the detection of *CT* (targeted genes: pmpH and cryptic plasmid). For *CT* serotyping, a LGV-specific probe (dark red) was chosen that hybridizes to the junction of a deletion area in the pmpH gene that is strictly present within LGV strains. The fluorescence signal of the LGV-specific probe is detected in channel 2. The non-LGV-specific probe (violet) hybridized within the deletion is in the pmpH gene, thus detecting *CT* serovars D-K, and the probe hybridized within the cryptic plasmid gene (yellow) detects all *CT* serovars. Both probes are labeled with the same fluorophore and are, thus, detected in the same channel (channel 3). The figure was created using BioRender.com. Abbreviations: F, fluorophore; Q, quencher.

Primer blast (NCBI, USA, selected database: nr) and the Oligo-Analyzer software (IDT, Coralville, IO, USA). To further prevent the formation of self- and cross-primer dimers, reverse and forward primers were modified with a 2′-O-methyl base in their penultimate base.

The temperature profile of the PCR is displayed in Table S7, and detailed information on the assembly of the master mix can be found in Table S8. The UC MMX-R2 (Utility Channel Master Mix Reagent 2, Roche, Rotkreuz, Switzerland) used in the master mix includes sequence-specific primers and probe for the amplification of the IC. The IC is spiked in automatically during extraction, thus acting as a build-in full-process control for each reaction. IC detection and the minimum of the relative fluorescence increase of corresponding curves can be customized using the respective software and is used as a threshold for the automatic calling of qualitative results.

## Evaluation of technical performance

The evaluation of the technical performance of the assay was conducted according to the new European Union regulations (Regulation 2017/746 EU IVDR). Quantitative reference standards were created using dPCR (Qiagen, Venlo, Netherlands) from clinical samples. Briefly, highly positive samples for *CT* serovars L1-L3 (LGV strain) and *NG* were selected and diluted to amount to a final volume of 2 mL. Nucleic acid extraction was performed on the QiaSymphony platform (Qiagen, Hilden, Germany). Next, serial dilutions of the clinical samples (10-fold, three steps, *n* = 1 repeat per dilution) were tested on a digital PCR system (Qiagen, Hilden, Germany) using the primers and probes described in this study [targeted genes for dPCR quantification: *CT*: pmpH (single-copy gene); *NG*: opa (multi-copy gene)]. The average of the concentrations in decimal copies per milliliter as determined by the corresponding digital PCR software was used for downstream experiments.

To determine the LLoD of the UC_CTNG assay in urine and swabs, the standards were used to create a twofold dilution series (eight steps, *n* = 21 repeats per dilution) on the Hamilton STARled IVD liquid handler (Hamilton Company, Bonaduz, Switzerland). The lowest applied concentration was 3.13 digital copies (dcp)/mL (swab)/3.91 dcp/mL (urine) and the highest 400 dcp/mL (swab)/1,000 dcp/mL (urine) for the assays detecting *CT* and 1.56 dcp/mL (swab)/3.91 dcp/mL (urine) and 200 dcp/mL (swab)/1,000 dcp/mL

(urine) for the assays detecting *NG*, respectively. cobas PCR and eSwab medium (ratio: 50:50) or cobas PCR medium and urine (ratio: 50:50), which tested negative for *CT* and *NG* by PCR, were used as the diluent. To examine whether swabs collected from different body sites interfere with the established technical performance of the assay, verification experiments using pooled clinical specimens (oropharyngeal, anal/rectal, cervical/vaginal, urethral/penile, and ocular swabs) as a matrix were used (diluted in cobas PCR media, ratio: 50:50). Matrix pools were tested non-reactive for *CT* and *NG* by PCR beforehand. Next, a twofold dilution series of the quantitative standards was tested for each matrix separately and in the eSwab medium (each six steps, eight repeats per step).

The assessment of linearity was performed by creating a 10-fold dilution series (four steps, *n* = 5 replicates per dilution) utilizing the standards quantified by dPCR. Inter- and intra-run variability was tested over the course of 3 days. On the first day, three samples (two positive and one negative, eSwab) were tested in a technical replica of three. On the following 2 days, the analysis was repeated with one replicate per sample. Exclusivity was assessed using a variety of different pathogens (*n* = 42, Table S3).

## Evaluation of clinical performance

The UC_CTNG assay was compared to an CE-IVD assay (cobas *CT*/*NG*, Roche, Rotkreuz, Switzerland) as a reference standard to assess assay performance. Sample analysis was performed on the same device and day. Both assays were subjected to a set of clinical samples, which included prospectively collected clinical specimens (*n* = 205, swabs and urine) and additional positive samples for *NG* and *CT,* which were identified by screening of our records (*n* = 114). Samples were stored at −20°C or lower until further use. For all swabs, the Copan eSwab collection and transport system (Copan Diagnostics Inc., Brescia, Italy) was used, and 1 mL of cobas PCR media (Roche, Rotkreuz, Switzerland) was added before testing. Urine samples were diluted 1:2 with cobas PCR media before testing.

For further evaluation of clinical performance, a set of EQA samples (*n* = 26) provided by INSTAND (Düsseldorf, Germany) containing either *NG*, *CT*, or both or neither pathogen was used. Next, for additional characterization of the assay, 7 months of testing under routine conditions was retrospectively analyzed (1 November 2022–31 May 2023). When possible, *NG* culture and PCR were performed in parallel. Briefly, a 10 µL sample (eSwab) was transferred to a selective agar for the isolation of *NG* and *N. meningitides* (chocolate agar PolyViteX VCAT3, BioMérieux, Marcy-l'Étoile, France) and incubated at 36 ± 1°C in 5% $CO_2$. Bacterial growth was assessed after 36–48 h according to our standard protocol. Suspected colonies were further analyzed by MALDI-TOF (MALDI Biotyper, Bruker, Billerica, MA, USA).

## Statistical analysis

Statistical analysis was conducted using GraphPad Prism version 9 (San Diego, CA, USA) and Validation Manager software (Finbiosoft, Espoo, Finland). For the analysis of the LLoD, a Probit analysis was computed. Figures 1 and 2 were created with BioRender.com. The use of anonymous samples was approved by the local ethics committee (Freie und Hansestadt Hamburg; No.: PV5626).

## ACKNOWLEDGMENTS

Marc Lütgehetmann and Holger Rohde are funded by the German Center of Infection Research (DZIF; institutional funding only).

We acknowledge financial support from the Open Access Publication Fund of UKE - Universitätsklinikum Hamburg-Eppendorf and DFG – German Research Foundation.

## AUTHOR AFFILIATIONS

[1]Center for Diagnostics, Institute of Medical Microbiology, Virology and Hygiene, University Medical Center Hamburg-Eppendorf (UKE), Hamburg, Germany
[2]German Center for Infection Research (DZIF), Hamburg-Lübeck-Borstel-Riems Site, Hamburg, Germany

## AUTHOR ORCIDs

Lisa Sophie Pflüger ⓘ http://orcid.org/0009-0009-1649-5312
Dominik Nörz ⓘ http://orcid.org/0000-0003-4001-7192
Susanne Pfefferle ⓘ http://orcid.org/0000-0001-7489-6557
Marc Lütgehetmann ⓘ http://orcid.org/0000-0002-9468-7944

## FUNDING

| Funder | Grant(s) | Author(s) |
|---|---|---|
| Deutsches Zentrum für Infektionsforschung | TTU 08.815 | Marc Lütgehetmann |
| | | Holger Rohde |

## AUTHOR CONTRIBUTIONS

Lisa Sophie Pflüger, Data curation, Formal analysis, Investigation, Visualization, Writing – original draft | Dominik Nörz, Conceptualization, Data curation, Formal analysis, Investigation, Supervision, Writing – review and editing | Moritz Grunwald, Data curation, Formal analysis | Susanne Pfefferle, Writing – review and editing | Katja Giersch, Data curation, Formal analysis, Writing – review and editing | Martin Christner, Writing – review and editing | Beatrice Weber, Writing – review and editing | Martin Aepfelbacher, Resources, Writing – review and editing | Holger Rohde, Writing – review and editing | Marc Lütgehetmann, Conceptualization, Data curation, Formal analysis, Resources, Supervision, Writing – review and editing

## ETHICS APPROVAL

The use of anonymous samples was approved by the local ethics committee (Freie und Hansestadt Hamburg; No.: PV5626).

## ADDITIONAL FILES

The following material is available online.

### Supplemental Material

**Supplemental material (Spectrum02756-23-s0001.docx).** Fig. S1 and Tables S1 to S8.

### Open Peer Review

**PEER REVIEW HISTORY (review-history.pdf).** An accounting of the reviewer comments and feedback.

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
