## [Reviewer comments · Microbiology Spectrum]

Microbiology Spectrum

Analytical and clinical validation of a multiplex PCR assay for detection of *Neisseria gonorrhoeae* and *Chlamydia trachomatis* including simultaneous LGV serotyping on an automated high-throughput qPCR system.

Lisa Sophie Pflüger, Dominik Nörz, Moritz Grunwald, Susanne Pfefferle, Katja Giersch, Martin Christner, Beatrice Weber, Martin Aepfelbacher, Holger Rhode, and Marc Lütgehetmann

Corresponding Author(s): Marc Lütgehetmann, University Medical Center Hamburg-Eppendorf

Review Timeline:

Submission Date:	August 1, 2023
Editorial Decision:	September 13, 2023
Revision Received:	November 10, 2023
Accepted:	January 8, 2024

Editor: Ana Cabrera

Reviewer(s): Disclosure of reviewer identity is with reference to reviewer comments included in decision letter(s). The following individuals involved in review of your submission have agreed to reveal their identity: Megan Amerson (Reviewer #2)

Transaction Report:

DOI: <https://doi.org/10.1128/spectrum.02756-23>

September 13, 2023

Dr. Marc Lütgehetmann
University Medical Center Hamburg-Eppendorf
Department of Microbiology, Virology and Hygiene
Hamburg
Germany

Re: Spectrum02756-23 (Analytical and clinical validation of a multiplex PCR assay for detection of *Neisseria gonorrhoeae* and *Chlamydia trachomatis* including simultaneous LGV serotyping on an automated high-throughput qPCR system.)

Dear Dr. Marc Lütgehetmann:

Thank you for submitting your manuscript to Spectrum. The manuscript was evaluated by two reviewers with expertise in your field. They have identified minor modifications, please address their comments below.

Link Not Available

Sincerely,

Ana Cabrera

Journals Department
Reviewer comments:

Reviewer #1 (Comments for the Author):

This is an excellent study describing the development of a sensitive *Neisseria gonorrhoeae* and *Chlamydia trachomatis* multiplex assay. Given that this assay was created from previously published primers/probes, I believe the work done by the authors to validate this assay was thorough. I just have a few comments/questions about the results.

- Line 91 - "my harbor fragments of" spelling error
- Line 96 - grammar error "The aim of this study was, to compile a multiplex"

- You define acronyms in the methods section, however that comes after the results, so I would recommend also defining the acronyms in the results section to avoid confusion
- Line 108 - I am confused by the number in the brackets next to the name of the primers, e.g (opa-rev (01)). You should refer to table 1 here to avoid confusion.
- Line 161 -Can you please explain this sentence: " For each assay one false positive and four false negative test results could be determined."
- Line 173 - "In 4,5 % of chlamydial ..." There is a typo
- Section 2.4 - I am a bit confused about the clinical samples tested. A total of 4,298 samples were screened. Of this, 1,914 were additionally tested by NG culture. Then, on line 177, you mention 1,914 culture negative samples, is this a mistake? Also, on line 178 you mention extra-genital specimens, are these part of the 4298 samples? I think some clarity is needed in this section.
- Line 201, you speak of another study, was this part of this study? Or unpublished data? I think clarification is needed.
- Line 294 - how was the standard created? I think more detail is needed for this section.

Reviewer #2 (Comments for the Author):

This is a well written analysis of a new laboratory developed test designed to identify *Chlamydia trachomatis* and *Neisseria gonorrhoeae*, in addition the assay is able to identify CT serovars associated with lymphogranuloma venereum.

Major revisions:

As matrix of the specimen type is important to indicate there should be more emphasis put on the different specimen types collected via swab. These specimen types should be evaluated independently as different specimen types and body fluids associated with those sites can differently impact extraction and amplification of the target. Example: ocular swabs and rectal swabs should be evaluated separately with adequate positive and negative samples from each sample type. Refer to CLSI and validation requirements for LDTs.

Was a cutoff Ct value defined for your assay for each target as the result of this study?

For *Neisseria* please define the criteria of a positive and negative value. Since 2 targets are involved do both need to be positive or one positive at a low Ct value (if so what is the cutoff). This is not clear.

Minor revisions:

Line 60: Capitalize "World Health Organization"

Line 66-70: multiple grammatical issues.

Line 105-106: reword for clarity. I think you mean that there is not a high risk of formation of primer-dimers; the sentence does not currently read as such.

Line 115: define dcp

Line 120-123: The swabs should not be lumped together for calculating the LLOD unless all of the swabs were obtained from the same body site. Different specimen sample types are likely to have different LLODs dependent upon body fluid composition at site. See major revision comment.

Line 169-170: Write out the dates. Do not use abbreviations.

Line 169: Specify how many samples came from each sample type/body site

Line 195: add spp after N.

Line 211: Please clarify if oropharyngeal specimens are hard to eradicate or hard to diagnose because of diagnostic assay limitations. If difficult to treat, please provide a reference.

Staff Comments:

Preparing Revision Guidelines

Please return the manuscript within 60 days; if you cannot complete the modification within this time period, please contact me. If you do not wish to modify the manuscript and prefer to submit it to another journal, please notify me of your decision immediately so that the manuscript may be formally withdrawn from consideration by Microbiology Spectrum.

Point-to-point response

Reviewer #1 (Comments for the Author):

This is an excellent study describing the development of a sensitive *Neisseria gonorrhoeae* and *Chlamydia trachomatis* multiplex assay. Given that this assay was created from previously published primers/probes, I believe the work done by the authors to validate this assay was thorough. I just have a few comments/questions about the results.

Response: We would like to thank the reviewer for the positive comments and time spend on reviewing the paper. The comments and corrections proposed by the reviewer certainly improved the manuscript.

- Line 91 - "my harbor fragments of" spelling error

Response: The spelling error was corrected (see line 91).

- Line 96 - grammar error "The aim of this study was, to compile a multiplex"

Response: The sentence was changed accordingly (see line 96).

- You define acronyms in the methods section, however that comes after the results, so I would recomme'nd also defining the acronyms in the results section to avoid confusion

Response: We fully agree with the reviewer on this point. The acronyms are now introduced in the first section they are mentioned in.

- Line 108 - I am confused by the number in the brackets next to the name of the primers, e.g (opa-rev (01)). You should refer to table 1 here to avoid confusion.

Response: For clarity, oligonucleotide names have been changes throughout the manuscript and in table 1 (e.g. from opa-rev (01) to opa-rev_01). Further, we added a note specifying that the oligonucleotide names in the manuscript match the names in table 1 (see lines 109 and 110).

- Line 161 - Can you please explain this sentence: " For each assay one false positive and four false negative test results could be determined."

Response: For both pathogens detected by the UC_CTNG assay, one false positive and four false negatives were identified when comparing test results to the comparator assay (CE-IVD, CTNG assay, Roche). False positive results were defined as UC_CTNG positive and CE-IVD negative, while false negative results were defined as UC_CTNG negative and CE-IVD positive. For this analysis, both *Neisseria gonorrhoeae* targets (opa and porA) must be reactive with the new UC_CTNG assay for the overall result to be considered *Neisseria gonorrhoeae* positive.

We share the reviewer's opinion that more clarity is needed in this section. The manuscript has been adapted to be more precise and a reference to figure 2 has been added (see lines 176 – 179).

- Line 173 - "In 4,5 % of chlamydial ..." There is a typo

Response: The typing error was corrected (see line 195).

- Section 2.4 - I am a bit confused about the clinical samples tested. A total of 4,298 samples were screened. Of this, 1,914 were additionally tested by *Neisseria gonorrhoeae* culture. Then, on line 177, you mention 1,914 culture negative samples, is this a mistake? Also, on line 178 you mention extra-genital specimens, are these part of the 4298 samples? I think some clarity is needed in this section.

Response: We thank the reviewer for bringing this mistake to our attention. Indeed, the numbers were incorrect. We screened 1,962/4,298 samples additionally by culture (not 1,914/4,298 as mentioned in the manuscript) of which 62 samples were positive for both the opa and the porA target with the UC_CTNG assay. The numbers have been changes accordingly and the sentence was rephrased for clarity (see lines 186 and 198-200).

Moreover, we have revised the statement regarding the extra-genital samples analyzed to specify that we are referring to the analysis of the oropharyngeal swabs that were screened as part of the evaluation of the assay performance in routine conditions (see lines 200 and 201).

- Line 201, you speak of another study, was this part of this study? Or unpublished data? I think clarification is needed.

Response: Again, we thank the reviewer for pointing out that more clarification is needed. We added the information that the retrospective analysis of the 4,298 clinical samples was part of the evaluation of the clinical performance of the new UC_CTNG assay of this study (see line 222).

- Line 294 - how was the standard created? I think more detail is needed for this section.

Response: Clinical samples that tested highly positive for *Chlamydia trachomatis* serovars L1-L3 (LGV strain) and *Neisseria gonorrhoeae* using our in-house LDT were selected. If necessary, the clinical samples were diluted to reach a final volume of 2 ml and aliquots containing 500 µl each were stored at -20° C. Next, one aliquot each was thawed and tested in a serial dilution series (10-fold, 3 steps) on a digital PCR Platform (Qiagen, Hilden, Germany) according to the manufacture's instructions. A negative control was included for each target. Nucleic acid extraction was performed on the QiaSymphony platform (Qiagen, Hilden, Germany). The same primers and probes as described in the study were used for amplification of the pmpH (*Chlamydia trachomatis*) and the opa gene (*Neisseria gonorrhoeae*). Determination of lower limit of detection and linearity were based on the average concentration as reported by the respective digital PCR software in digital copies/ml of the clinical samples. For each experiment a new separate aliquot was used to assure that the same conditions applied for all tests.

It is important to note, that for each pathogen detected with the UC_CTNG assay only one target was quantified. However, the opa target (*Neisseria gonorrhoeae*) is a multi-copy gene whereas the porA target is only a single-copy target. We thank the reviewer for the pointing out that more detail was needed about the quantification of the standards, as we feel that this clarification improved the manuscript (see updated methods section lines 328 – 335 and legend of table 2).

Reviewer #2 (Comments for the Author):

This is a well written analysis of a new laboratory developed test designed to identify *Chlamydia trachomatis* and *Neisseria gonorrhoeae*, in addition the assay is able to identify CT serovars associated with lymphogranuloma venereum.

Response: We appreciate the reviewer's careful evaluation of the manuscript and their helpful and detailed feedback, which has enhanced the quality of the paper.

Major revisions:

- As matrix of the specimen type is important to indicate there should be more emphasis put on the different specimen types collected via swab. These specimen types should be evaluated independently as different specimen types and body fluids associated with those sites can differently impact extraction and amplification of the target. Example: ocular swabs and rectal swabs should be evaluated separately with adequate positive and negative samples from each sample type. Refer to CLSI and validation requirements for LDTs.

Response: We agree with the reviewer that different matrixes can impact extraction and amplification. Therefore, careful evaluation and validation is required especially concerning different clinically relevant matrixes before using newly established assays in routine diagnostics.

To address this topic, we have performed further experiments to verify that lower limits of detection (LLoDs) are similar in swabs taken from different body sites. For this purpose, we have collected swabs from the most clinically relevant body sites at our center (oropharyngeal, anal/rectal, cervical/vaginal, urethral/penile and ocular). All swabs used for these verification experiments were collected and stored in eSwab medium (Copan). 3-4 matrix-pools per specimen type each containing 9-12 ml were created and tested for *Chlamydia trachomatis* and *Neisseria gonorrhoeae* by qPCR. Only non-reactive pools were included for further verification experiments. All matrix-pools were diluted 1:2 with Cobas PCR media (CPM, Roche). In addition to the matrix-pools also eSwab media diluted with CPM (1:2) was tested in the same manner to establish the technical performance without the presence of possible PCR inhibitors from human specimens. Next, dilution series from the same clinical standards used to determine LLoDs in eSwab and urine were used. The standards were stored in aliquots at -20°C (freeze-thaw-cycles: 0). Serial dilutions were designed to start well above the beforehand established LLoDs in eSwab and urine. The highest applied concentration was 800 dcp/ml and the lowest 25 dcp/ml. All concentrations were tested in technical replicas of 8.

LLoDs were determined to be similar compared to those established in eSwab media for oropharyngeal and ocular swabs and slightly higher for anal/rectal and urethral/penile swabs. However, cervical/vaginal swabs performed significantly worse compared to all other matrixes tested. This could be partly due to mucus that is present in cervical and vaginal swabs and known to interfere with PCR performance. Still, all internal control test results were valid for each dilution tested (for all tested matrixes). The detailed results of the verification experiments were added as a new supplementary table (supplementary table 1 and an explanation was added to the manuscript (see lines 126-133 and 343-349). Further, the results are enclosed in this point-to-point response at the end of this comment.

To provide a more comprehensive overview of the 4,298 clinical samples that were tested as part of this study under routine laboratory conditions, we have added a new supplementary table showing a detailed analysis of all swabs that were tested during the respective seven-month long period. The table lists all swabs tested according to the body sites where the swabs were collected and the respective number of *Chlamydia trachomatis* and *Neisseria gonorrhoeae* positive samples.

Another important point is to ensure a high quality of newly established LDTs, is to prevent false-negative test result due to PCR inhibition. To this end, we chose the option "positive target confirmation with IC (internal control)" for the UC_CTNG assay. The IC is thus automatically spiked-in before extraction, thereby acting as a full process control. If the IC is not detected in the beforehand selected range (between ct min. and max), the PCR test result is marked as invalid.

Moreover, we would like to point out that regulations and guidelines on requirements for the use of LDTs for routine diagnostic vary depending on the different respective regulatory bodies as well as from country to country. Therefore, it is important to note, that the in our study described establishment and validation of the UC_CTNG assay warrants further verification experiments if use of the assay for diagnostics purposes is planned, especially concerning the different matrixes that the assay is going to be used for. This point was also added in discussion (see line 255-261).

**Technical performance in swabs collected from different body sites**

| | | *Chlamydia trachomatis* | | | | *Neisseria gonorrhoeae* | | | |
|---|---|---|---|---|---|---|---|---|---|
| | | serovars A-L3* | | serovars L1-L3* | | opa traget** | | porA** | |
| **swab type** | conc. [dcp/ml] | positive results | hitrate | positive results | hitrate | positive results | hitrate | positive results | hitrate |
| **oropharyngeal swabs** | 800 | 8/8 | 100 % | 8/8 | 100 % | 8/8 | 100 % | 8/8 | 100 % |
| | 400 | 8/8 | 100 % | 8/8 | 100 % | 8/8 | 100 % | 8/8 | 100 % |
| | 200 | 8/8 | 100 % | 8/8 | 100 % | 8/8 | 100 % | 7/8 | 87.5 % |
| | 100 | 8/8 | 100 % | 8/8 | 100 % | 8/8 | 100 % | 6/8 | 75 % |
| | 50 | 8/8 | 100 % | 8/8 | 100 % | 8/8 | 100 % | 1/8 | 12.5 % |
| | 25 | 8/8 | 100 % | 7/8 | 87.5 % | 8/8 | 100 % | 3/8 | 37.5 % |
| **anal/rectal swabs** | 800 | 8/8 | 100 % | 8/8 | 100 % | 8/8 | 100 % | 8/8 | 100 % |
| | 400 | 8/8 | 100 % | 8/8 | 100 % | 8/8 | 100 % | 6/8 | 75 % |
| | 200 | 8/8 | 100 % | 8/8 | 100 % | 8/8 | 100 % | 6/8 | 75 % |
| | 100 | 8/8 | 100 % | 8/8 | 100 % | 6/8 | 75 % | 4/8 | 50 % |
| | 50 | 6/8 | 75 % | 6/8 | 75 % | 3/8 | 37.5 % | 2/8 | 25 % |
| | 25 | 4/8 | 50 % | 4/8 | 50 % | 1/8 | 12.5 % | 1/8 | 12.5 % |
| **urethral/penile swabs** | 800 | 8/8 | 100 % | 8/8 | 100 % | 8/8 | 100 % | 8/8 | 100 % |
| | 400 | 8/8 | 100 % | 8/8 | 100 % | 8/8 | 100 % | 4/8 | 50 % |
| | 200 | 8/8 | 100 % | 8/8 | 100 % | 8/8 | 100 % | 4/8 | 50 % |
| | 100 | 8/8 | 100 % | 8/8 | 100 % | 8/8 | 100 % | 5/8 | 62.5 % |
| | 50 | 7/8 | 87.5 % | 8/8 | 100 % | 4/8 | 50 % | 2/8 | 25 % |
| | 25 | 1/8 | 12.5 % | 6/8 | 75 % | 4/8 | 50 % | 1/8 | 12.5 % |
| **cervical/vaginal swabs** | 800 | 7/8 | 87.5 % | 5/8 | 62.5 % | 6/8 | 75 % | 1/8 | 12.5 % |
| | 400 | 7/8 | 87.5 % | 5/8 | 62.5 % | 4/8 | 50 % | 1/8 | 12.5 % |
| | 200 | 2/8 | 25 % | 1/8 | 12.5 % | 4/8 | 50 % | 0/8 | 0 % |
| | 100 | 2/8 | 25 % | 3/8 | 37.5 % | 2/8 | 25 % | 1/8 | 0 % |
| | 50 | 3/8 | 37.5 % | 0/8 | 0 % | 0/8 | 0 % | 0/8 | 0 % |
| | 25 | 1/8 | 12.5 % | 1/8 | 12.5 % | 2/8 | 25 % | 0/8 | 0 % |
| **ocular swabs** | 800 | 8/8 | 100 % | 8/8 | 100 % | 8/8 | 100 % | 8/8 | 100 % |
| | 400 | 8/8 | 100 % | 8/8 | 100 % | 8/8 | 100 % | 8/8 | 100 % |
| | 200 | 8/8 | 100 % | 8/8 | 100 % | 8/8 | 100 % | 7/8 | 87.5 % |
| | 100 | 8/8 | 100 % | 8/8 | 100 % | 8/8 | 100 % | 5/8 | 62.5 % |
| | 50 | 8/8 | 100 % | 8/8 | 100 % | 8/8 | 100 % | 4/8 | 50 % |
| | 25 | 8/8 | 100 % | 8/8 | 100 % | 8/8 | 100 % | 4/8 | 50 % |
| **eSwab medium (Copan)** | 800 | 8/8 | 100 % | 8/8 | 100 % | 8/8 | 100 % | 8/8 | 100 % |
| | 400 | 8/8 | 100 % | 8/8 | 100 % | 8/8 | 100 % | 8/8 | 100 % |
| | 200 | 8/8 | 100 % | 8/8 | 100 % | 8/8 | 100 % | 6/8 | 75 % |
| | 100 | 8/8 | 100 % | 8/8 | 100 % | 8/8 | 100 % | 4/8 | 50 % |
| | 50 | 8/8 | 100 % | 8/8 | 100 % | 8/8 | 100 % | 3/8 | 37.5 % |
| | 25 | 7/8 | 87.5 % | 6/8 | 75 % | 3/8 | 100 % | 0/8 | 0 % |

- Was a cutoff Ct value defined for your assay for each target as the result of this study?

Response: No cutoff ct values were defined during this study. When using the open channel (cobas Utility channel) to establish laboratory developed tests on the cobas5800/6800/8800, the respective software defines positive and negative results automatically based on the selected minimal relative fluorescence increase (RFI min.). Generally, the RFI is defined as the plateau fluorescence level/baseline fluorescence level. Using the Utility channel software, the RFI min. for automated calling of test results can be selected during programming of the analysis package for each channel separately. The selected RFIs min. for the UC_CTNG assay are listed in **supplementary table 7**. To determine the adequate RFI min for each channel, a set of positives and negatives is tested using the respective primer/probe sets and the amplification curves are analyzed using the Utility channel software. If the baseline or plateau fluorescence level are not suitable to select a reliable RFI min., probes are modified accordingly (e.g. using internal quenchers). In addition to the automated calling of test results according to the selected RFIs min., the software uses an algorithm to check whether amplification curves are sigmoidal.

To further ensure a high quality of the new qPCR assay, we chose the option “positive target confirmation with IC (internal control)” during analysis package installation. The IC is spiked in automatically during nucleic acid extraction and sequence-specific primers and probe for amplification and detection of the IC in channel 5 are included in the reagents provided by the manufacturer (MMRX-2, Roche). In addition to the RFI min. (which is set at 2 for channel 5 as recommended by the manufacturer), min. and max. ct values can be selected to determine the range in which the software will accept the IC as valid. Subsequently, the IC acts as a full process control and samples are marked as invalid if the requirements for the IC are not met.

As an additional quality control measurement for diagnostics at our laboratory, a low and high positive control of known concentrations as well as a negative control are used in each run. If the ct values of the respective positive controls differs < 0.5 ct of the expected value, a new positive control is thawed and the run is repeated until all controls yield valid results.

- For Neisseria please define the criteria of a positive and negative value. Since 2 targets are involved do both need to be positive or one positive at a low Ct value (if so what is the cutoff). This is not clear.

Response: The UC_CTNG assay is considered *Neisseria gonorrhoeae* positive if both targets (opa and porA) yield a positive test result. The software automatically classifies results as positive or negative for each channel separately depending on the selected relative fluorescence increase min. (see explanation above). If only one of the *Neisseria gonorrhoeae* targets tests positive (opa+/porA- or opa-/porA+) the test has to be repeated or a confirmation by culture is warranted to distinguish between commensal *Neisseria* species, *Neisseria meningitides* and *Neisseria gonorrhoeae*.

We agree with the reviewer that a more detailed explanation of the interpretation of the *Neisseria gonorrhoeae* test results is needed in the manuscript. Therefore, we made appropriate additions to the results and methods sections (see lines 153-156). Further we added a second reference to figure 2 that depicts the interpretation of test results for each possible combination of test results (see lines 155 and 156).

Minor revisions:

- Line 60: Capitalize "World Health Organization"

Response: This spelling error was corrected (see lines 60 and 61).

- Line 66-70: multiple grammatical issues.

Response: The sentences were rephrased (see lines 65-69).

- Line 105-106: reword for clarity. I think you mean that there is not a high risk of formation of primer-dimers; the sentence does not currently read as such.

Response: The sentence was rephrased (see lines 106 and 107).

- Line 115: define dcp

Response: The acronym stands for digital copies which indicated that the copy number of the respective pathogen was determined using a digital PCR platform. The explanation of the term has been introduced at its first mention in the manuscript (see lines 117 and 118).

- Line 120-123: The swabs should not be lumped together for calculating the LLOD unless all of the swabs were obtained from the same body site. Different specimen sample types are likely to have different LLODs dependent upon body fluid composition at site. See major revision comment.

Response: Lower limits of detection (LLODs) were obtained using two highly positive clinical samples (*Chlamydia trachomatis* serovar L1-L3 and *Neisseria gonorrhoeae*) which were quantified using digital PCR (see last comment from reviewer number 1 for a more detailed explanation on how the standards were created). To assess LLODs in different matrices, both clinical standards were added to either eSwab medium or urine (pooled urine samples, tested negative for *Chlamydia trachomatis* and *Neisseria gonorrhoeae* by qPCR) mixed with cobas PCR media (ratio: 50:50) and tested in serial dilutions (2-fold, 8 steps, 21 repeats/step). To further evaluate LLODs in swabs taken from different body sites (all collected and stored in eSwab medium), verification experiments were performed as part of this revision. For further details on how the verification experiments were performed and the respective results see response to the first major revision comment of reviewer number 2.

- Line 169-170: Write out the dates. Do not use abbreviations.

Response: The requested change was made in the manuscript (see line 189).

- Line 169: Specify how many samples came from each sample type/body site

Response: In total 4,298 samples were screened which included predominantly swabs (n=2,876) and urine samples (n=1,391) as well as a small number of assorted samples (n=31). For all swabs analyzed, the eSwab collection and transport system was used (Copan Diagnostics Inc., Brescia, Italy). The majority of swabs were oropharyngeal (n=1,174) and anal/rectal (n=1,085) swabs but the cohort included also urethral/penile (n=325), vaginal/cervical (n=250) and ocular swabs (n=26) as well as 16 swabs that were not properly labelled (no collection site indicated or unreadable). The information has been added in the manuscript (see lines 187 and 188).

- Line 195: add spp after N.

Response: The requested change was made in the manuscript (see line 218).

- Line 211: Please clarify if oropharyngeal specimens are hard to eradicate or hard to diagnose because of diagnostic assay limitations. If difficult to treat, please provide a reference.

Response: We agree with the reviewer that some clarification concerning this fact is needed. The sentence was changed to be more clear and point out the fact that because oropharyngeal *Neisseria gonorrhoeae* infections are often a- or paucisymptomatic these infections are regularly missed as clinicians do not order testing for *Neisseria gonorrhoeae* (see lines 235 and 236).

Re: Spectrum02756-23R1 (Analytical and clinical validation of a multiplex PCR assay for detection of *Neisseria gonorrhoeae* and *Chlamydia trachomatis* including simultaneous LGV serotyping on an automated high-throughput qPCR system.)

Dear Dr. Marc Lütgehetmann:

Thank you for submitting your revised manuscript and for addressing all reviewer comments.

Your manuscript has been accepted, and I am forwarding it to the ASM production staff for publication. Your paper will first be checked to make sure all elements meet the technical requirements. ASM staff will contact you if anything needs to be revised before copyediting and production can begin. Otherwise, you will be notified when your proofs are ready to be viewed.

Sincerely,
Ana Cabrera
Editor
Microbiology Spectrum

Reviewer #2 (Comments for the Author):

This is a well written analysis of a new laboratory developed test designed to identify *Chlamydia trachomatis* and *Neisseria gonorrhoeae*, in addition the assay is able to identify CT serovars associated with lymphogranuloma venereum.

No additional comments or suggestions. All previous reviewer comments have been adequately addressed